# Effect of Nano-Phosphorus Formulation on Growth, Yield and Nutritional Quality of Wheat under Semi-Arid Climate

Anuj Poudel [1], Satish Kumar Singh [1,*], Raimundo Jiménez-Ballesta [2], Surendra Singh Jatav [1,*], Abhik Patra [1,3] and Astha Pandey [1]

1   Department of Soil Science and Agricultural Chemistry, Institute of Agricultural Sciences, Banaras Hindu University, Varanasi 221005, Uttar Pradesh, India
2   Department of Geology and Geochemistry, Autónoma University of Madrid, 28049 Madrid, Spain
3   Krishi Vigyan Kendra, Narkatiaganj, West Champaran 845455, Bihar, India
*   Correspondence: sksingh_1965@rediffmail.com (S.K.S.); surendra.jatav1@bhu.ac.in (S.S.J.)

**Abstract:** Appropriate phosphorus (P) management techniques increase yield and nutritional properties while minimizing environmental concerns. The widespread use of nano-fertilizers (NFs) in agriculture endangers soil and plants. It is vital to research the behavior of nano-phosphors (nano-P) on plant growth and quality, as well as their technique of interaction with soil properties in order to obtain key ecosystem benefits. With this in mind, a field experiment was conducted using wheat as a test crop to explore the impact of nano phosphorus (nano-P) on soil. The study's goal was to examine how the foliar application of nano-P to wheat affects its growth, yield and nutrient concentration. Treatments consisted of: $T_1$: 100% NPK (120:137:72 kg N:$P_2O_5$:$K_2O$ ha$^{-1}$) by RDF (recommended dose of fertilizer); $T_2$: 100% NPK by RDF + 2 foliar sprays of nano-P @ 494.21 mL ha$^{-1}$; $T_3$: 100% NK + 0% P (no foliar); $T_4$: 100% NK + 75% P + 2 foliar sprays of nano-P @ 494.21 mL ha$^{-1}$; $T_5$: 100% NK + 50% P + 2 foliar sprays of nano-P @ 494.21 mL ha$^{-1}$; $T_6$: 100% NK + 0% P + 2 foliar sprays of nano-P @ 494.21 mL ha$^{-1}$; $T_7$: 100% NPK by RDF + 1 foliar spray of nano-P @ 494.21 mL ha$^{-1}$; $T_8$: 100% NK + 75% P + 1 foliar spray of nano-P @ 494.21 mL ha$^{-1}$; $T_9$: 100% NPK + 1 foliar spray of nano-P @ 494.21 mL ha$^{-1}$; $T_{10}$-100% NK + 75% P + 1 foliar spray of nano-P @ 494.21 mL ha$^{-1}$. According to the findings, applying 100% NK + 75% P + 2 foliar applications of nano-P at the tillering and panicle initiation stages increased yield over 100% RDF by 37.1%. Additionally, the highest micronutrient concentration (Zn (36.4 mg kg$^{-1}$), Cu (21.2 mg kg$^{-1}$), Mn (22.9 mg kg$^{-1}$) and Fe (61.1 mg kg$^{-1}$)) in grain were noticed in $T_3$ (100% NK + 0% P no foliar spray of nano-P) treatment, which was superior to $T_1$ (100% NPK). Furthermore, foliar application of nano-P fertilizer in combination with different levels of diammonium phosphate (DAP) slightly increased the amount of N, P and K, as well as micronutrients in post-harvest soil. In summary, the use of 100% NK + 75% + 2 foliar applications of nano-P saved 25% recommendation dose P if supplied as nano-P as a form of phosphorus, and can be a suitable substitute for DAP, especially in smart agriculture, as it possibly reduces P leaching into groundwater, while maintaining or increasing wheat crop yield over the 100% RDF.

**Keywords:** foliar application; diammonium phosphate; nano-phosphorus; micronutrients; wheat

## 1. Introduction

Cereal crops serve a critical role in fulfilling the worldwide demand for food of an extensively increasing population, particularly in underdeveloped countries where agriculture-based on cereals are the primary nutrition and calorie sources [1,2]. One of the major important cereal crops of India is wheat and it plays an important role in economy, as well as the food security of the country. In the year 2020/2021, wheat production during the Rabi season in India was over 107 MMT from 31.6 million hectares, and shared around 37% of total food grain production [3]. Wheat productivity has increased dramatically after the green revolution, owing to the increasing use of plant protection chemicals and irrigated

areas. However, the unbalanced and indiscriminate use of these chemical fertilizers had a negative impact on soil health, human health and factor production [4,5]. Fertilization management is currently one of the most challenging tasks under the field management. Despite the fact that mineral fertilizers are necessary for the growth of plants, their prolonged usage poses environmental and health risks, such as surface and groundwater pollution. However, the fertilizer industry in India is strictly monitored and regulated by the Indian government. After nitrogen (N), P has been identified as the most deficient critical nutrient in many agricultural production systems [6]. Wheat requires P for development from its seedling stage to maturity. This nutrient helps to ensure consistent heading, faster maturity, winter hardiness seed development, root development, tiller formation, and grain filling [6]. Phosphorus helps in photosynthesis, energy storage, and cell division from the background. Between the financial year of 2020 and 2021, the output of DAP, the second most popular fertilizer among Indian farmers after urea, climbed steadily at 1.9%, compared to an impressive 8% growth between FYs 2019 and 2020. The decrease in growth percentage was caused by a scarcity of raw materials and a rise in input prices, particularly rock phosphate. India, the world's largest importer of urea, is a key customer of diammonium phosphate (DAP)needed to feed the country's massive agriculture industry, which employs around 60% of the workforce and accounts for 15% of the $ 2.7 trillion economy of the country.

All of these constraints can be overcome by using smart delivery systems, such as nano-technology and nano-fertilizers, that can assist long-term soil health and agricultural output [7]. Nano-fertilizers exhibit greater nutrient use efficiency due to their improved capacity to penetrate and translocate within plant parts. Additionally, by avoiding interaction between nutrients and soil, water, air and microbes, it can achieve direct internalization by crops, which limits unwanted nutrient loss [8,9]. Nano-fertilizers are fertilizers that contain nutrients inside nano porous materials covered with polymer films, or given as nano-scale emulsions or particles [10]. Nano-fertilizers regulate the nutrient release depending on the crop requirement, making them more efficient than normal fertilizers [11]. The increase in P concentration after the application of nano-P may be because diameters of 25–50 nm help retain P as a result of increased total surface area and protect P from fixation resulting in control release of nutrients, making P available for a longer time due to increased concentration of P. An increased level of available N, along with P and K, in plants as well as soil after harvesting, was reported with increasing foliar application of nano-P fertilizer [12]. With the increase of available nutrients, the uptake of the nutrients, as well as NUE, is increased in wheat crops and increases the yield. An indiscriminate application of P fertilizers results in eutrophication, as they enter the aquatic environment via leakage, leaching and runoff [13]. Furthermore, P is a finite, non-renewable resource. Within the next 50–100 years, P reserves are supposed to be depleted according to certain studies [14,15]. Phosphorus applied by foliar application has greater utilization efficiency than P supplied directly to the soil, P administered via foliar method may improve soil applied P, hence increasing P usage efficiency and reducing crop reliance on soil P. Nano-P formulations can reduce nutrient losses through direct internalization of crops, while synthetic P fertilizers have high fixation rates in the soil and low uptake efficiency. Phosphorus applied in the form of nano-fertilizers (NFs) can be an excellent alternative, especially in modern agriculture systems, as it has a slow release material over a long period of time, reducing P leaching into underground water and promoting sustainable productivity and quality. Nano-fertilizers have a nutrient use efficiency (NUE) of 58–51%, whereas SSP and DAP have NUEs of 15–16%. Nano-P costs USD 4.29–4.82 per acre, depending on the leaf size of the plants, whereas SSP costs USD 5.85–7.80 and DAP costs USD 18–24. The newly created nano-fertilizer will reduce chemical fertilizer consumption 80–100 times, saving significant foreign exchange on fertilizer imports. In India, nano-DAP was recently introduced by Indian Farmers Fertilizer Cooperative Limited (IFFCO) to meet the demands of farmers. Field studies to assess the relative potential of diammonium phosphate sources (DAP and nano-DAP) began in *Kharif* 2021 and have yielded extremely promising findings.

It is necessary to critically examine the scientific understanding of several elements influencing the accessibility of indigenous and applied P to crops. The use of NFs instead of conventional synthetic fertilizers is a way to release nutrients in a controlled and conditional way, thus reducing the loss of nutrients, soil toxicity and maintaining sustainability and protection of agriculturally produced food [16,17]. The nano-P, or their aggregates, enter the cuticle easily and directly through the cuticular pathway, as well as move long distances in the plant vascular system through the stomatal pathway [18]. Foliar application, according to Arif [19], is a viable strategy for increasing the availability of nutrients to crops in order to boost output. They came to the conclusion that foliar sprays of nutrient solutions at various growth stages, together with recommended amounts of fertilizers, and the application of nano-DAP, resulted in improved wheat yield, yield components, biomass and pronounced P content, even for 75% lower input than commercial DAP [20]. On applying nano-hydroxyapatite fertilizers on Panicum, Reis et al. [21] found that the maximum supply of nutrients via nano-composites is better matched to the demands of plants, resulting in higher P-use efficiency [22]. Meena et al. [23] reported an increase in grain and straw yield of wheat by 44.6 and 13.1% after the application of NFs, which may be attributed to the increased growth hormone, enhanced metabolic process and photosynthetic activities. The efficacy of soil fertilizer application is lower than that of foliar fertilization under various environmental conditions, due to the direct provision of necessary nutrients to the leaves, relatively quicker absorption, independence of root activity, and soil water availability. Sarkar et al., [24] reported that application of NCPC-H (nano-phosphatic fertilizer) considerably improved P uptake (32.4 mg pot$^{-1}$) by pearl millet. As a result, when used in conjunction with foliar spray of nano-P, it aids in reducing the recommended P levels. Low rates of foliar sprayed nano-P may address mid-season P deficit in winter wheat, resulting in greater P-use efficiency when compared to soil applications. Nasrallah et al. [25] observed that total yield, along with different parameters of plant growth of broad bean, was improved by 30% when treated with calcium phosphate nano-particles as compared to conventional means of fertilizers, which may be accredited to better nutrient uptake and augmentation in total soluble sugars. Application of nano-hydroxyapatite (nHAP) and nano-P used as P sources in acidic, as well as basic soils, revealed enhanced overall germination, P content, biomass, and plant length in tomatoes [26]. On using calcium phosphate nano-particles, Elsayed et al. [27] observed different growth characteristics and physiological indices of rosemary, compared to the traditional method of fertilizer application.

The novelty of the present study is the application of nano-P as phosphorus fertilizer, along with different levels of DAP for sustainable agricultural. Nano-P is a new liquid fertilizer, and it could be substituted for DAP. Because the rate of fixation of soil-applied P fertilizers is very high, the unit cost of fertilizer is also very high. Therefore, there is a need for effective application of P fertilizers, offering scope for evaluating nano-P fertilizers. The current study was conducted (i) to determine the effect of nano-P fertilizers added as foliar application on growth, yield and yield components, as well as the chemical composition of grain of wheat under semi-arid climate.

## 2. Materials and Methods

### 2.1. Study Area

The experiment was conducted on the Agriculture Research Farm, Institute of Agricultural Sciences, Banaras Hindu University, Varanasi, Uttar Pradesh, India, during the *Rabi* season of 2020–21. The experimental site is located at 25°26′ N latitude and 82°99′ E longitude, at an altitude of 128.93 m above mean sea-level in the north Gangetic plains (Figure 1). The climate of the area of the experimental site falls in a semi-arid to sub-humid climate, characterized by hot summers and cold winters. The initial soil had a pH (1:2.5 in water) of 7.3, EC of 0.29 dS m$^{-1}$, organic carbon (OC) of 0.34%, available N of 93 mg kg$^{-1}$, available P of 11 mg kg$^{-1}$, and available K of 66 mg kg$^{-1}$. The DTPA-extractable Cu, Mn, Zn, and Fe contents in soil were 2.09, 2.33, 0.53 and 2.34 mg kg$^{-1}$, respectively. The



soil of the experimental field was sandy loam in texture, corresponding to the USDA Soil Taxonomy. The main soil type was Inceptisol.

**Figure 1.** Location of the experimental site, layout and experimental view.

## 2.2. Experimental Design and Treatments

The field experiment was carried out in a randomized complete block design (RCBD), taking ten treatments with three replications in the plot size of 5 m × 3 m (15 m$^2$). The experiment consisted of treatments comprising NPK or NK, along with foliar spray of nano–P combinations (Table 1), applied to HD-2967 variety of winter wheat. The sowing wheat was done in the first week of December, with a row-to-row spacing of 20 × 20 cm, and was harvested in the second week of April the following year. The recommended dose of fertilizer (RDF) for wheat was 120:137:72 kg N:P$_2$O$_5$:K$_2$O ha$^{-1}$ kg N:P$_2$O$_5$:K$_2$O ha$^{-1}$. N (at half dose) and P$_2$O$_5$ and K$_2$O (at their full) were applied at the time of sowing by the means of urea, di-ammonium phosphate, and muriate of potash, respectively. The remaining half dose of N was used in two equal splits at tillering and panicle/ear initiation stages of the crop. However, the first application of foliar sprays of nano-P was given at the tillering stage and second application at the panicle initiation stage in selected treatments. Nano-P was applied at the rate of 494.21 mL ha$^{-1}$. Calculation was done accordingly for the experimental plot size.

**Table 1.** Treatment details.

| Symbols | Treatment Details | Time of Nano-P Application |
|---|---|---|
| T$_1$ | 100% NPK by RDF (Recommended dose of fertilizer) | (No foliar spray of nano-P) |
| T$_2$ | 100% NPK by RDF + 2 foliar sprays of nano-P @ 494.21 mL ha$^{-1}$ | Tillering and panicle initiation stage |
| T$_3$ | 100% NK + 0% P (No foliar) | |
| T$_4$ | 100% NK + 75% P + 2 foliar sprays of nano-P @ 494.21 mL ha$^{-1}$ | Tillering and panicle initiation stage |
| T$_5$ | 100% NK + 50% P + 2 foliar sprays of nano-P @ 494.21 mL ha$^{-1}$ | Tillering and panicle initiation stage |
| T$_6$ | 100% NK + 0% P + 2 foliar sprays of nano-P @ 494.21 mL ha$^{-1}$ | Tillering and panicle initiation stage |
| T$_7$ | 100% NPK by RDF + 1 foliar spray of nano-P @ 494.21 mL ha$^{-1}$ | Tillering stage |
| T$_8$ | 100% NK + 75% P + 1 foliar spray of nano-P @ 494.21 mL ha$^{-1}$ | Tillering stage |
| T$_9$ | 100% NPK + 1 foliar spray of nano-P @ 494.21 mL ha$^{-1}$ | Panicle initiation stage |
| T$_{10}$ | 100% NK + 75% P + 1 foliar spray of nano-P @ 494.21 mL ha$^{-1}$ | Panicle initiation stage |

### 2.3. Yield and Yield Attributes of Wheat

Various yield attributes of wheat, such as panicle length and the number of grains panicle$^{-1}$, were recorded at maturity. After cutting off the panicles, plants were harvested, leaving 5 cm from the base to avoid soil contamination. The plants were then washed with 0.2% detergent solution, 0.1 N HCl, and finally twice with distilled water before being dried in a hot air oven at 60 °C until a constant weight was reached. The dried plant samples collected from 1 m$^2$ area were threshed to separate the grains from the straw. The threshed grain and straw were weighted using a weighing balance, then converted to t ha$^{-1}$ grain and straw yield. On the other hand, the biological yield was calculated by adding both the grain and straw yield and 1000 grain weight counted (test wight), and the harvest index (HI) was calculated using the following formula [5]:

$$\text{Harvest Index} = \frac{\text{Grain yield}\left(\text{kg ha}^{-1}\right)}{\text{Biological yield}\left(\text{kg ha}^{-1}\right)} \times 100$$

### 2.4. Chemical Analysis of Soil and Plants

After the crop harvest, soil samples from the depth of 0–15 cm were taken from each plot and air-dried before being transferred through a 2 mm sieve. Soil samples were analyzed for the determination of pH and electrical conductivity (EC) (1:2.5) using a glass electrode pH meter and digital EC meter, available nitrogen (N) by using the Kjeldahl autoanalyzer (DISTYL-EM; Pelican) [28], available phosphorus (P) by spectrophotometer [29], available potassium (K) by a flame photometer [30] and Cu, Mn, Zn, Fe were determined by using an atomic absorption spectrophotometer (AAS), as per the standard procedure [31]. The plant material was dried at 60 °C for 72 h in a hot air oven. Dry plant tissue was finely grounded in a soil-processing lab and stored in zipped polythene bags. The nitrogen concentration was determined by digestion ($H_2SO_4$), distillation and titrimetric method, using a standard Kjeldahl autoanalyzer (DISTYL-EM; Pelican) procedure. Grain and straw samples were digested in a di-acid mixture ($HNO_3$:$HClO_4$:3:1 *v/v*) for the estimation of cationic micronutrients (Fe, Cu, Zn and Mn), by AAS (Agilent FS-240) [32].

### 2.5. Statistical Analysis

To test for statistically significant differences among the ten treatments, a one-way analysis of variance (one-way ANOVA) was performed using the SPSS Statistics 20.0 (SPSS Inc., Chicago, IL, USA). The Duncan multiple range test (DMRT), at <0.05 levels of significance, was used to evaluate the significant differences among the mean values. The Pearson correlation analysis among parameters was performed using the R-square (R version 3.5.1).

## 3. Results and Discussion

### 3.1. Yield and Yield Parameters

Results depicted in Table 2 shows that the application of foliar NP significantly increased the panicle number plant$^{-1}$, panicle length (cm) and 1000 seed weight (g). The highest values of 5.67, 12.4 cm and 41.1 were recorded with treatment $T_4$ (100% NK + 75%P + 2 foliar sprays of nano-P at the tillering and panicle initiation stages), which resulted an increase of 35.5, 34.7 and 29.15%, respectively, over $T_1$. It is well known that applying foliar application of nano-fertilizer with fertilizer to croplands can enhance yield. Sirisena et al. [33] found similar result in rice with nano-fertilizer application. This might be due to the fact that the addition of nano-P with DAP enhances the direct availability of N and P from chemical fertilizers and nano-P, which results in increased leaf area and higher dry matter accumulation. The use of nano-P improves plant metabolic processes and photosynthesis, as a result increasing the number of panicles and grain development, thus increasing wheat output and growth metrics [23]. Another reason may be that the application nano-P improves nutrient absorption, resulting in optimal growth of plant parts and improved metabolic

processes such as photosynthesis, which results in higher accumulation of photosynthates and translocation to the plant's economic parts, improving crop growth, development and yield [34]. A similar result was seen on broad bean, where the use of nano-fertilizer provided better nutrient accumulation and increased growth activity due to smart delivery system of the fertilizers [12].

**Table 2.** Effects of nano-P on yield attributes and yield of wheat.

| Treatments | Number of Panicles Plant$^{-1}$ | Panicle Length (cm) | Grain Yield (t ha$^{-1}$) | Straw Yield (t ha$^{-1}$) | Test Weight (g) | Biological Yield (t ha$^{-1}$) | Harvest Index (%) |
|---|---|---|---|---|---|---|---|
| T1 | 4.33 e | 9.2 c | 2.89 d | 3.67 c | 31.8 e | 6.56 c | 44.2 bc |
| T2 | 5.44 abc | 10.7 b | 3.77 ab | 4.17 ab | 36.2 c | 7.93 b | 47.5 a |
| T3 | 3.89 e | 9.0 c | 2.50 e | 3.48 c | 32.2 e | 5.98 c | 41.8 a |
| T4 | 5.67 a | 12.4 a | 3.97 a | 4.77 a | 41.1 a | 8.73 a | 45.4 ab |
| T5 | 4.89 d | 10.7 b | 3.32 c | 4.47 ab | 35.8 c | 7.78 b | 42.6 bc |
| T6 | 4.33 e | 9.68 c | 3.25 cd | 4.48 ab | 34.2 d | 7.73 b | 42.0 a |
| T7 | 5.56 ab | 12.19 a | 3.78 ab | 4.58 ab | 40.8 a | 8.37 ab | 45.2 ab |
| T8 | 5.00 cd | 10.89 b | 3.52 bc | 4.20 b | 36.1 c | 7.7.2 b | 45.5 ab |
| T9 | 5.44 abc | 12.1 a | 3.57 bc | 4.40 ab | 38.3 b | 7.97 b | 44.8 abc |
| T10 | 5.11 bcd | 10.4 b | 3.33 c | 4.30 ab | 36.6 c | 7.63 b | 43.7 bc |
| SEM± | 0.15 | 0.17 | 1.25 | 1.57 | 0.38 | 2.15 | 1.31 |
| CD at 5% | 0.46 | 0.5 | 3.74 | 4.67 | 1.13 | 6.4 | NS |

Different letters indicate significant differences at the 5% level according to a Duncan's test. Mean (±SE) was taken from three replicates for each treatment; CD: Critical deference.

Nano-P application had a substantial impact on both grain and straw yields of wheat, according to critical data analysis shown in Table 2. It is noticeable that the use of nano-P, in conjunction with varying amounts of P fertilizer, has a substantial impact on grain and straw yield. Grain and straw yields considerably enhanced in treatment $T_4$ to 3.97 t ha$^{-1}$ and 4.77 t ha$^{-1}$, respectively, representing a significant increase of 37.1 and 29.9% above RDF ($T_1$) alone, followed by increases of 30.8 and 24.9% in treatment $T_7$. Adhikari et al. [35] speculated that the yield of grain and straw increment over 100 RDF due to better absorption, interception and utilization of P in the nano-P form as P is released slowly throughout the growth period, resulting in a higher photosynthetic rate and ultimately, a higher biomass yield accumulation. Liu and Lal [36] reported that the application of nano-particle fertilizer resulted in increased growth rate by 32.6% and seed production by 20.4% in comparison to no nano-fertilizer. Our findings of this study corroborate previous studies, in which foliar spray of nano-fertilizer boosted wheat crop yield and yield attributes [37,38].

From the experimental, data it was found that grain and straw yield of wheat considerably increased or decreased over 100% RDF ($T_1$) (Figure 2). Application of a reduced dose of P fertilizer, i.e., 0, 50, and 75% of RDF, along with single- or double-spray of nano-P, i.e., $T_5$ (100% NK + 50% P + 2 foliar sprays of nano-P), $T_6$ (100% NK + 0% P + 2 foliar sprays of nano-P), $T_8$ (100% NK + 75% P + 1 foliar spray of nano-P), $T_{10}$ (100% NK + 75% P + 1 foliar spray of nano-P), recorded 14.7, 12.3, 21.5 and 15.2% increase in wheat grain yield over $T_1$, respectively. However, the decrement was noticed in $T_3$ (13.6%) over RDF. In regards to straw yield, the highest increase in straw production over 100% RDF ($T_1$) was recorded in 100% NK + 75% P + 2 foliar sprays of nano-P at the tillering and panicle initiation stages ($T_4$; 29.9%), followed by 100% NPK by RDF + 1 foliar spray of nano-P at the tillering stage ($T_7$, 24.9%), 100% NK + 0% P + 2 foliar sprays of nano-P ($T_6$, 22.25%) and 100% NK + 50% P + 2 foliar sprays of nano-P ($T_5$, 21.8%). Similarly, decrement in straw yield was reported in $T_3$ (5.0%). In comparison to soil application, the foliar application of nano-P at the tillering and panicle initiation stages, that is directly involved in the metalloprotease and enzymatic activities in plants that are important for plant growth and

development, increased the grain and straw yield percentage to more than 100% RDF, and nano-P provided targeted delivery of nutrients throughout the crop growth period, aligning with the results of [35].

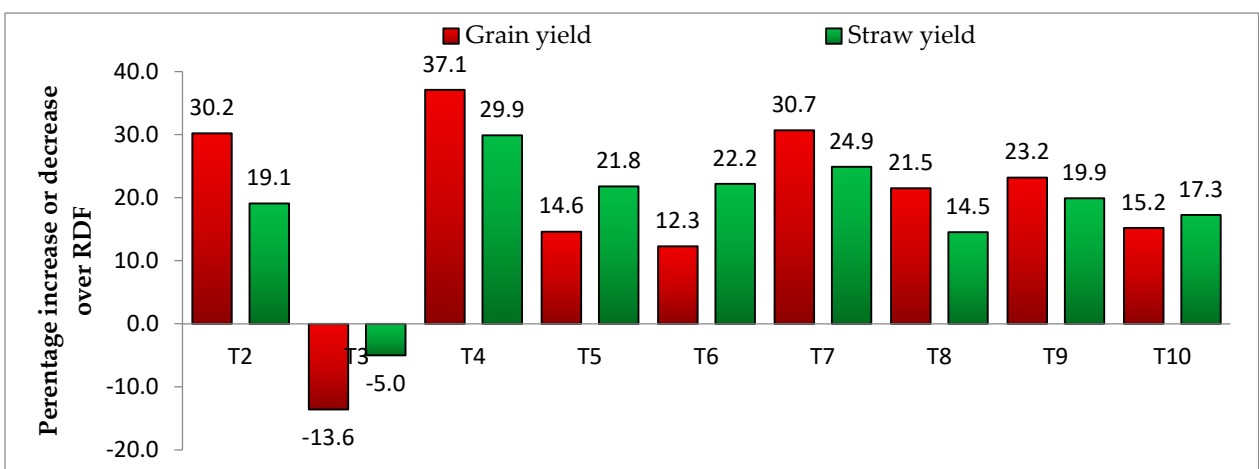

**Figure 2.** Effect of nano-P on percent increase or decrease in grain and straw yield of wheat ($T_1$ = 100%).

### 3.2. Effect of Nano-Phosphorus Fertilizers on the Nutrient Concentration

The findings in Table 3 show that varied dosages of nano- and inorganic P fertilizer had a considerable influence on nutrient concentration in wheat grain and straw. The maximum value of N concentration in grain (1.68%) and straw (0.57%) was achieved from treatment $T_4$, which corresponded to a considerable increase of 36.5 and 29.5% over RDF. Similarly, foliar nano-P treatment enhanced P content in both grain and straw. Treatment $T_4$ had the greatest concentrations of P in grain (0.29%) and straw (0.14%), which increased by 70.5 and 52.6% in comparison to RDF, respectively. Similarly, the maximum K concentration in grain and straw was recorded in treatment $T_4$, which was 0.47 and 1.45%, respectively, with a substantial increase of 23.6 and 20.8% above RDF.

**Table 3.** Effects of nano-P on nitrogen, phosphorus and potassium concentration (%) in the grain and straw of wheat.

| Treatments | Nitrogen (%) | | Phosphorus (%) | | Potassium (%) | |
|---|---|---|---|---|---|---|
| | Grain | Straw | Grain | Straw | Grain | Straw |
| $T_1$ | 1.23 [e] | 0.44 [d] | 0.172 [f] | 0.095 [ef] | 0.38 [de] | 1.30 [d] |
| $T_2$ | 1.48 [c] | 0.55 [ab] | 0.235 [d] | 0.138 [ab] | 0.41 [cd] | 1.36 [bc] |
| $T_3$ | 1.40 [cd] | 0.43 [d] | 0.134 [g] | 0.088 [f] | 0.36 [e] | 1.29 [d] |
| $T_4$ | 1.68 [a] | 0.57 [a] | 0.290 [a] | 0.145 [a] | 0.47 [a] | 1.45 [a] |
| $T_5$ | 1.39 [cd] | 0.49 [c] | 0.188 [e] | 0.117 [cd] | 0.39 [de] | 1.34 [cd] |
| $T_6$ | 1.35 [d] | 0.49 [c] | 0.167 [f] | 0.111 [de] | 0.38 [de] | 1.33 [cd] |
| $T_7$ | 1.59 [b] | 0.57 [a] | 0.274 [b] | 0.136 [ab] | 0.45 [ab] | 1.41 [ab] |
| $T_8$ | 1.45 [c] | 0.51 [c] | 0.201 [e] | 0.132 [abc] | 0.42 [bcd] | 1.36 [bc] |
| $T_9$ | 1.47 [c] | 0.52 [bc] | 0.260 [bc] | 0.128 [bc] | 0.44 [abc] | 1.40 [b] |
| $T_{10}$ | 1.45 [c] | 0.51 [c] | 0.257 [c] | 0.131 [abc] | 0.41 [cd] | 1.39 [b] |
| SEM± | 0.03 | 0.01 | 0.005 | 0.004 | 0.009 | 0.014 |
| CD at 5% | 0.1 | 0.03 | 0.015 | 0.012 | 0.027 | 0.042 |

Different letters indicate significant differences at the 5% level according to the Duncan's test. Mean (±SE) was taken from three replicates for each treatment; CD: Critical deference.

Furthermore, the application of nano-P, along with different amounts of P fertilizer, has affected the micronutrient concentration in the grain and straw of wheat, as depicted in Table 4. Treatment $T_3$ had the maximum Cu, Mn, Zn and Fe concentrations in grain, i.e., 21.2, 22.9, 36.4 and 61.1 mg kg$^{-1}$, respectively; whereas, in straw, the same treatment had the highest Cu, Mn, Zn and Fe concentrations, i.e., 15.0, 13.4, 25.3 and 25.1 mg kg$^{-1}$, respectively. Szameitat et al. [39] discovered that applying nano-hydroxyapatite (nHAP) increased P concentration and, as the result of orthophosphates released from dissolved nHAP, allowed full functionality restoration of P in treated plants. Nano-particles enter the plant system by interacting with ionic channels, transport proteins, aquaporins, forming new pores, or as a result of endocytosis, all of which result in higher nutrient concentrations in rice [40]. According to Kaviani et al., [41], foliar-applied nano-P had a substantially favorable influence on leaf N, P and K concentrations in plants that were treated compared to control plants. A similar finding was noticed in our study, but the highest micronutrient concentration was noticed in the treatment that did not receive the P, which may imply that the availability of P can reduce micronutrient availability due to the negative interaction between micronutrients and P in the soil system. Hussien et al. [42] concur with these findings, as they discovered that applying nano-P at 1.0 g L$^{-1}$ resulted in the highest concentrations of nutrients in cotton plant leaves. Hanif et al. [43] found that the entry of nano-particles into plants can drive metabolic activities, resulting in a higher rate of exudation, which favors higher micronutrient concentrations. Dhansil et al. [44] reported that the application of NF helped in increasing the phosphorus content in straw and grain of pearl millet. Wheat plants sprayed with a combination of NFs and amino acid gave the highest values of Zn concentration (0.926 and 0.891 mg kg$^{-1}$) in grain [45–47].

**Table 4.** Effects of nano-P on micronutrient concentration (mg kg$^{-1}$) in the grain and straw of wheat.

| Treatments | Cu (mg kg$^{-1}$) | | Mn (mg kg$^{-1}$) | | Zn (mg kg$^{-1}$) | | Fe (mg kg$^{-1}$) | |
|---|---|---|---|---|---|---|---|---|
| | Grain | Straw | Grain | Straw | Grain | Straw | Grain | Straw |
| $T_1$ | 11.7 [f] | 12.2 [de] | 19.4 [cd] | 11.2 [b] | 19.8 [e] | 12.3 [f] | 54.4 [c] | 21.7 [b] |
| $T_2$ | 12.4 [f] | 14.3 [ab] | 21.8 [ab] | 11.1 [b] | 23.3 [d] | 14.7 [de] | 56.1 [bc] | 22.7 [b] |
| $T_3$ | 21.2 [a] | 15.0 [a] | 22.9 [a] | 13.4 [a] | 36.4 [a] | 25.3 [a] | 61.1 [a] | 25.1 [a] |
| $T_4$ | 18.5 [b] | 12.3 [cde] | 20.7 [bc] | 11.1 [b] | 27.8 [b] | 16.3 [cd] | 57.3 [b] | 23.9 [ab] |
| $T_5$ | 17.0 [cd] | 13.2 [bcd] | 16.1 [e] | 11.9 [b] | 25.8 [c] | 17.0 [c] | 56.2 [bc] | 23.4 [ab] |
| $T_6$ | 15.4 [e] | 14.3 [ab] | 16.9 [e] | 12.2 [b] | 26.0 [c] | 21.1 [b] | 56.2 [bc] | 22.9 [b] |
| $T_7$ | 17.3 [bc] | 11.5 [d] | 20.6 [bc] | 11.3 [b] | 25.2 [c] | 17.1 [c] | 56.3 [bc] | 22.6 [b] |
| $T_8$ | 15.8 [de] | 13.6 [bc] | 21.8 [ab] | 11.6 [b] | 23.5 [d] | 14.2 [e] | 54.3 [c] | 23.5 [ab] |
| $T_9$ | 17.8 [bc] | 11.8 [de] | 17.1 [e] | 11.1 [b] | 25.0 [c] | 16.1 [cd] | 54.7 [bc] | 22.1 [b] |
| $T_{10}$ | 18.1 [bc] | 11.9 [de] | 18.7 [d] | 11.5 [b] | 23.7 [d] | 15.8 [cde] | 55.0 [bc] | 22.7 [b] |
| SEM± | 0.52 | 0.32 | 0.58 | 0.37 | 0.34 | 0.49 | 0.75 | 0.65 |
| CD at 5% | 1.55 | 0.94 | 1.74 | 1.1 | 1.01 | 1.45 | 2.24 | 1.92 |

Different letters indicate significant differences at the 5% level according to the Duncan's test. Mean (±SE) was taken from three replicates for each treatment; CD: Critical deference.

*3.3. Effect of Nano-Phosphorus Fertilizers in the % Increase or Decrease of P Concentration in the Grain and Straw of Wheat*

Figure 3 shows that the P content in wheat grain and straw rose or reduced significantly when compared to 100% RDF ($T_1$). The application of 100% NK + 75% P + 2 foliar sprays of nano-P at the tillering and panicle initiation stages ($T_4$) resulted in the highest P content in grain (71.4%), followed by 100% RDF ($T_1$) + 1 foliar spray of nano-P at the tillering stage ($T_7$) (58.4%).Treatment $T_5$ (100 % NK + 50% P + 2 foliar sprays of nano-P), T8 (100% NK + 75% P + 1 foliar spray of nano-P), and $T_{10}$ (100% NK + 75% P + 1 foliar spray of nano-P) observed 9.9, 16.2, and 53.1% increases in grain P content over $T_1$. However, there was a decrease in $T_3$ (20.5 %) and $T_6$ (2.8%) over 100% RDF. In terms of P concentration in straw, the greatest increase over 100% RDF ($T_1$) was observed in 100% NK + 75% P + 2 foliar sprays of nano-P at the tillering and panicle initiation stages

(T4; 51.7%), followed by 100% NPK + 2 foliar sprays of nano-P (T$_2$; 45.3%) and 100% NK + 75% P + 1 foliar spray of nano-P (T$_{10}$; 40.4%). Similarly, a decrease in the P content of straw (5.0%) was noted as compared to RDF. Dhansil et al. [44] reported that the application of nano fertilizers helped in increasing the phosphorus content in straw and grain of pearl millet. The phosphorus content increased in straw from 0.12 to 0.25%, and in grain from 0.24 to 0.44%. Similarly, the supply of P as nano-KH$_2$PO$_4$ promoted greater physiological efficiency of the shoots and roots for P, resulting in increased P concentration in the shoots and roots, which may be due to increased photosynthetic rate [40].

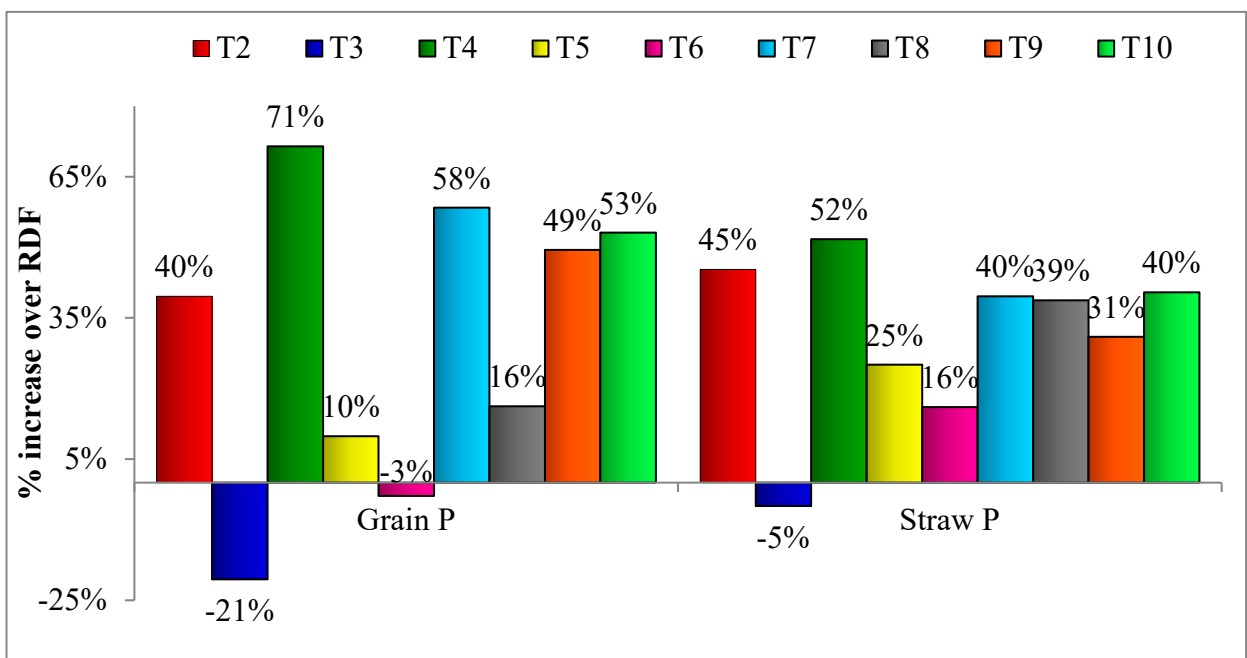

**Figure 3.** Effects of nano-P fertilizers in the % increase or decrease of P concentration in the grain and straw of wheat.

### 3.4. Nutrients Uptake

Based on the result of different levels of nano-P fertilization, a convincing effect on the uptake of macronutrient was obtained (Table 5). Wheat grain N, P, and K intake varied from 35.7 to 66.6, 3.37 to 11.5, and 9.98 to 18.4 kg ha$^{-1}$, respectively. The absorption of macronutrients (N, P, and K) in straw ranged from 14.9 to 27.1, 3.08 to 6.92, and 44.9 to 68.9 kg ha$^{-1}$, respectively. Treatment T$_4$ had significantly increased total N uptake (81.6% higher than RDF). Similarly, treatment T$_4$ had higher total P absorption (117% over RDF), while treatment T$_4$ had higher total K uptake (49.5% over RDF). All nano-P treatments were most effective in total macronutrient absorption by wheat in T$_4$ (100% NK + 75% P + 2 foliar sprays of nano-P at the tillering and panicle initiation stages). Total Cu, Mn, Zn, and Fe uptake ranged from 78.7 to 132, 101 to 135, 109 to 179, and 240 to 341 g ha$^{-1}$, as shown in Table 6. The treatments T$_4$ and T$_6$ had significantly higher grain and straw uptake of Cu (73.5 and 65.8 g ha$^{-1}$), Mn (82.1 and 55.7 g ha$^{-1}$), and Zn (98.5 and 85.6 g ha$^{-1}$); whereas, T$_4$ had the highest grain and straw uptake of Fe (227 and 114 g ha$^{-1}$). According to Soliman et al., [48], there is a positive relationship between P and N, indicating that as P intake increases, so does N uptake. The NCPC addition (NCPC-H) reported a 42% higher P uptake by pearl millet over CF (CF-H) (CF-H) [20]. Phosphorus uptake increased in broad bean by 6.7 and 5.24% in the straw and grain, respectively, after the application of nano-P fertilizer [12]. Kandil and Marie [45] also reported that the combination of NF and amino acid sprayed on wheat plants had the maximum micronutrient uptake in grain and straw. Use of NF increased the micronutrient concentration in the roots and shoots of lettuce plants, which is reflected in plant growth [49].

**Table 5.** Effects of nano-P on nitrogen, phosphorus and potassium uptake.

| Treatments | Nitrogen (kg ha$^{-1}$) | | | Phosphorus (kg ha$^{-1}$) | | | Potassium (kg ha$^{-1}$) | | |
|---|---|---|---|---|---|---|---|---|---|
| | Grain | Straw | Total | Grain | Straw | Total | Grain | Straw | Total |
| T$_1$ | 35.7 [f] | 16.0 [c] | 51.8 [f] | 5.04 [f] | 3.48 [g] | 8.52 [g] | 11.0 [f] | 47.4 [c] | 58.5 [c] |
| T$_2$ | 55.7 [bc] | 22.9 [b] | 78.6 [bc] | 8.85 [c] | 6.02 [bc] | 14.9 [c] | 15.5 [bc] | 56.9 [b] | 72.4 [b] |
| T$_3$ | 35.0 [f] | 14.9 [c] | 49.9 [f] | 3.37 [g] | 3.08 [f] | 6.45 [h] | 8.98 [g] | 44.9 [c] | 53.9 [c] |
| T$_4$ | 66.6 [a] | 27.1 [a] | 93.8 [a] | 11.5 [a] | 6.92 [a] | 18.4 [a] | 18.4 [a] | 68.9 [a] | 87.3 [a] |
| T$_5$ | 46.1 [de] | 21.7 [b] | 67.8 [e] | 6.24 [de] | 5.23 [de] | 11.5 [e] | 13.0 [def] | 59.8 [b] | 72.8 [b] |
| T$_6$ | 44.0 [e] | 22.2 [b] | 66.2 [e] | 5.46 [ef] | 4.96 [e] | 10.4 [f] | 12.5 [ef] | 61.2 [b] | 73.6 [b] |
| T$_7$ | 60.0 [b] | 25.5 [a] | 85.6 [b] | 10.4 [b] | 6.22 [a] | 16.6 [b] | 17.0 [ab] | 63.3 [ab] | 80.4 [ab] |
| T$_8$ | 50.9 [cd] | 21.5 [b] | 72.4 [cde] | 7.02 [d] | 5.53 [d] | 12.5 [d] | 14.8 [cd] | 57.3 [b] | 72.2 [b] |
| T$_9$ | 52.6 [cd] | 23.0 [b] | 75.6 [cd] | 9.27 [c] | 5.64 [cd] | 14.9 [c] | 15.5 [bc] | 61.5 [c] | 77.0 [b] |
| T$_{10}$ | 48.2 [de] | 22.1 [b] | 70.3 [cd] | 8.60 [c] | 5.63 [cd] | 14.2 [c] | 13.6 [cde] | 59.7 [b] | 73.3 [b] |
| SEM± | 2.13 | 0.77 | 2.28 | 0.35 | 0.21 | 0.31 | 0.67 | 2.26 | 2.52 |
| CD at 5% | 6.33 | 2.29 | 6.78 | 1.06 | 0.64 | 0.94 | 1.99 | 6.72 | 7.50 |

Different letters indicate significant differences at the 5% level according to the Duncan's test. Mean (±SE) was taken from three replicates for each treatment; CD: Critical deference.

**Table 6.** Effects of nano-P on micronutrient uptake (g ha$^{-1}$) in wheat.

| Treatments | Copper (g ha$^{-1}$) | | | Manganese (g ha$^{-1}$) | | | Zinc (g ha$^{-1}$) | | | Iron (g ha$^{-1}$) | | |
|---|---|---|---|---|---|---|---|---|---|---|---|---|
| | Grain | Straw | Total | Grain | Straw | Total | Grain | Straw | Total | Grain | Straw | Total |
| T$_1$ | 33.7 [f] | 45.0 [c] | 78.7 [c] | 60.0 [b] | 41.1 [c] | 101 [b] | 64.2 [e] | 45.0 [f] | 109 [f] | 168 [de] | 80.0 [d] | 248 [d] |
| T$_2$ | 46.7 [e] | 59.7 [ab] | 106 [b] | 82.1 [a] | 46.2 [bc] | 128 [a] | 88.0 [abcd] | 61.3 [e] | 149 [de] | 212 [ab] | 94.7 [bcd] | 306 [bc] |
| T$_3$ | 52.9 [de] | 52.4 [bc] | 105 [b] | 57.2 [b] | 46.8 [bc] | 104 [b] | 91.1 [abc] | 81.1 [ab] | 172 [abc] | 153 [e] | 87.3 [cd] | 240 [d] |
| T$_4$ | 73.5 [a] | 58.8 [ab] | 132 [a] | 82.1 [a] | 53.0 [ab] | 135 [a] | 98.5 [a] | 77.5 [bc] | 176 [a] | 227 [a] | 114 [a] | 341 [a] |
| T$_5$ | 56.3 [cd] | 58.7 [ab] | 115 [b] | 53.6 [b] | 53.2 [ab] | 107 [b] | 85.4 [bcd] | 75.8 [bc] | 161 [bcd] | 187 [cd] | 105 [abc] | 291 [bc] |
| T$_6$ | 50.2 [de] | 65.8 [a] | 116 [b] | 55.0 [b] | 55.7 [a] | 111 [b] | 84.6 [bcd] | 85.6 [a] | 170 [abc] | 183 [cd] | 105 [abc] | 288 [bc] |
| T$_7$ | 65.4 [b] | 51.5 [bc] | 117 [b] | 78.2 [a] | 51.0 [ab] | 129 [a] | 95.5 [ab] | 78.4 [b] | 174 [ab] | 213 [ab] | 101 [abc] | 314 [b] |
| T$_8$ | 55.4 [cd] | 57.1 [ab] | 113 [b] | 76.8 [a] | 48.8 [ab] | 126 [a] | 82.3 [cd] | 59.6 [e] | 142 [e] | 191 [c] | 98.8 [abc] | 290 [bc] |
| T$_9$ | 63.6 [b] | 51.8 [bc] | 115 [b] | 60.9 [b] | 48.9 [ab] | 110 [b] | 89.3 [abcd] | 70.7 [cd] | 160 [cd] | 195 [bc] | 97.3 [bc] | 292 [bc] |
| T$_{10}$ | 60.4 [bc] | 51 [bc] | 111 [b] | 62.3 [b] | 49.4 [ab] | 112 [b] | 79.1 [d] | 67.9 [d] | 147 [e] | 183 [cd] | 97.6 [bc] | 281 [c] |
| SEM± | 2.46 | 2.64 | 3.58 | 3.79 | 2.35 | 4.60 | 3.53 | 2.90 | 4.93 | 7.49 | 5.0 | 10.1 |
| CD at 5% | 7.31 | 7.86 | 10.64 | 11.26 | 6.98 | 13.69 | 10.49 | 8.62 | 14.65 | 22.28 | 14.8 | 30.04 |

Different letters indicate significant differences at the 5% level according to the Duncan's test. Mean (±SE) was taken from three replicates for each treatment; CD: Critical deference.

### 3.5. Post-Harvest Soil Properties

The data pertaining to post-harvest properties of soil has been presented in Table 7. The highest soil pH (7.67) in post-harvest soil (PHS) was recorded in T$_4$ (100% NK + 75% P + 2 foliar sprays of nano-P @ 494.21 mL ha$^{-1}$) and the minimum in T$_3$ (100% NK + 0% P no foliar). Soil application P, along with foliar-application of nano-P treatments, resulted in a significantly increased soil pH when compared to no soil P application, but there was no significant difference when compared to 100% recommended dose of P. There was no significant change noticed in EC as compared to 100% RDF, except treatment in T$_4$ (100% NK + 75% P + 2 foliar sprays of nano-P @ 494.21 mL/ha$^{-1}$) and T$_3$ (100% NK + 0% P no foliar). However, treatment T$_4$ resulted in a significant increment, whereas in treatment T$_3$, a significant reduction was noticed as compared to the 100% RDF (T$_1$). The available

N content increased from 213.25 kg ha$^{-1}$ in RDF, to 234.15 kg ha$^{-1}$ in treatment T$_4$ (100% NK + 75% P + 2 foliar sprays of nano-P at the tillering and panicle initiation stages). Similarly, P content in post-harvest soil was affected by different levels of P fertilizers. The maximum available P content increased to 33.60 kg ha$^{-1}$ in treatment T$_7$ (100% NPK by RDF + 1 foliar spray of nano-P at the tillering stage) compared to 28.39 kg ha$^{-1}$ in RDF. The maximum available K content of 161.18 kg ha$^{-1}$ was recorded in treatment T$_4$ (100% NK + 75% P + 2 foliar sprays of nano-P at the tillering and panicle initiation stages) compared to 151.14 kg ha$^{-1}$ in RDF. The use of different DAP doses and foliar sprays of nano-P as a fertilizer sources resulted in a considerable change in the micronutrient content in post-harvest soil. The maximum Cu content of 1.78 mg kg$^{-1}$ was reported in treatment T$_6$ (100% NK + 0% P + 2 foliar sprays of nano-P at the tillering and panicle initiation stages), while the highest Mn content of 2.92 mg kg$^{-1}$ was reported by treatment T$_5$ (100% NK + 50% P + 2 foliar sprays of nano-P at the tillering and panicle initiation stages). Treatment deficit of P fertilizer showed an increase in Zn content, as P and Zn have an antagonistic relationship with each other. The maximum Zn content, i.e., 0.65 mg kg$^{-1}$, was reported in treatment T$_3$ (100% NK + 0% P no foliar spray of nano-P). Similarly, the highest Fe content (2.43 mg kg$^{-1}$) was also reported in treatment T$_3$ (100% NK + 0% P no foliar spray of nano-P). This could be because NF raised the concentration of nutrients in soil solution, resulting in higher osmotic potential and a little reduction in nutrient uptake, therefore nutrient retention in soil after crop harvesting [50]. Foliar treatment of nano-hydroxyapatite satisfied the P requirement of plants at each step in the soil cycle, and spraying nHA boosted nutrient status in soil after crop harvesting [48]. There was a significant increase in available macronutrients (N, P and K) in soil after harvesting with increasing foliar application of nano-fertilizers on broad bean [12].

**Table 7.** Effects of nano-P on pH, EC, nitrogen, phosphorus, potassium and DTPA extractable micronutrient content in the post-harvest soil.

| Treatments | pH | EC | Available N (kg ha$^{-1}$) | Available P (kg ha$^{-1}$) | Available K (kg ha$^{-1}$) | DTPA-Extractable Micronutrients (mg kg$^{-1}$) | | | |
|---|---|---|---|---|---|---|---|---|---|
| | | | | | | Cu | Mn | Zn | Fe |
| T$_1$ | 7.58 [ab] | 0.34 [bc] | 213 [abcd] | 28.3 [ab] | 151 [cd] | 1.43 [g] | 2.84 [b] | 0.54 [b] | 2.21 [g] |
| T$_2$ | 7.61 [ab] | 0.35 [abc] | 222 [abcd] | 32.5 [ab] | 154 [bcd] | 1.45 [g] | 2.64 [d] | 0.56 [b] | 2.25 [f] |
| T$_3$ | 7.22 [c] | 0.29 [d] | 201 [d] | 19.3 [d] | 148 [d] | 1.67 [c] | 2.53 [e] | 0.65 [a] | 2.43 [a] |
| T$_4$ | 7.67 [a] | 0.38 [a] | 234 [a] | 30.9 [ab] | 161 [a] | 1.52 [f] | 2.80 [b] | 0.54 [b] | 2.33 [de] |
| T$_5$ | 7.50 [b] | 0.33 [bc] | 209 [bcd] | 27.4 [bc] | 153 [abc] | 1.61 [d] | 2.92 [a] | 0.56 [b] | 2.37 [c] |
| T$_6$ | 7.29 [c] | 0.32 [bc] | 205 [cd] | 22.7 [cd] | 150 [d] | 1.78 [a] | 2.26 [f] | 0.62 [a] | 2.40 [b] |
| T$_7$ | 7.51 [b] | 0.36 [ab] | 230 [ab] | 33.6 [a] | 160 [ab] | 1.71 [b] | 2.79 [bc] | 0.55 [b] | 2.35 [cd] |
| T$_8$ | 7.51 [b] | 0.35 [abc] | 225 [abc] | 31.3 [ab] | 159 [ab] | 1.56 [e] | 2.72 [c] | 0.56 [b] | 2.32 [de] |
| T$_9$ | 7.59 [ab] | 0.37 [ab] | 226 [abc] | 32.9 [ab] | 157 [abc] | 1.53 [ef] | 2.79 [bc] | 0.54 [b] | 2.27 [f] |
| T$_{10}$ | 7.58 [ab] | 0.36 [ab] | 217 [abcd] | 31.9 [ab] | 159 [ab] | 1.65 [c] | 2.80 [b] | 0.55 [b] | 2.32 [de] |
| SEM± | 0.029 | 0.012 | 6.95 | 1.01 | 1.78 | 0.02 | 0.026 | 0.01 | 0.008 |
| CD at 5% | 0.09 | 0.04 | 20.66 | 3.01 | 5.28 | 0.008 | 0.08 | 0.03 | 0.02 |

Different letters indicate significant differences at the 5% level according to the Duncan's test. Mean (±SE) was taken from three replicates for each treatment; CD: Critical deference.

### 3.6. Correlation among Variables

In present study, soil pH ($p < 0.05$) at 5% significance level showed a significantly negative linear relationship with copper ($R^2 = -0.69$), zinc ($R^2 = -0.83$) and iron ($R^2 = -0.71$), while EC ($R^2 = 0.72$), nitrogen ($R^2 = 0.53$), phosphorus ($R^2 = 0.80$), potassium ($R^2 = 0.50$), manganese ($R^2 = 0.68$), grain yield ($R^2 = 0.63$) and straw yield ($R^2 = 0.31$) exhibited a significantly positive correlation (Figure 4).

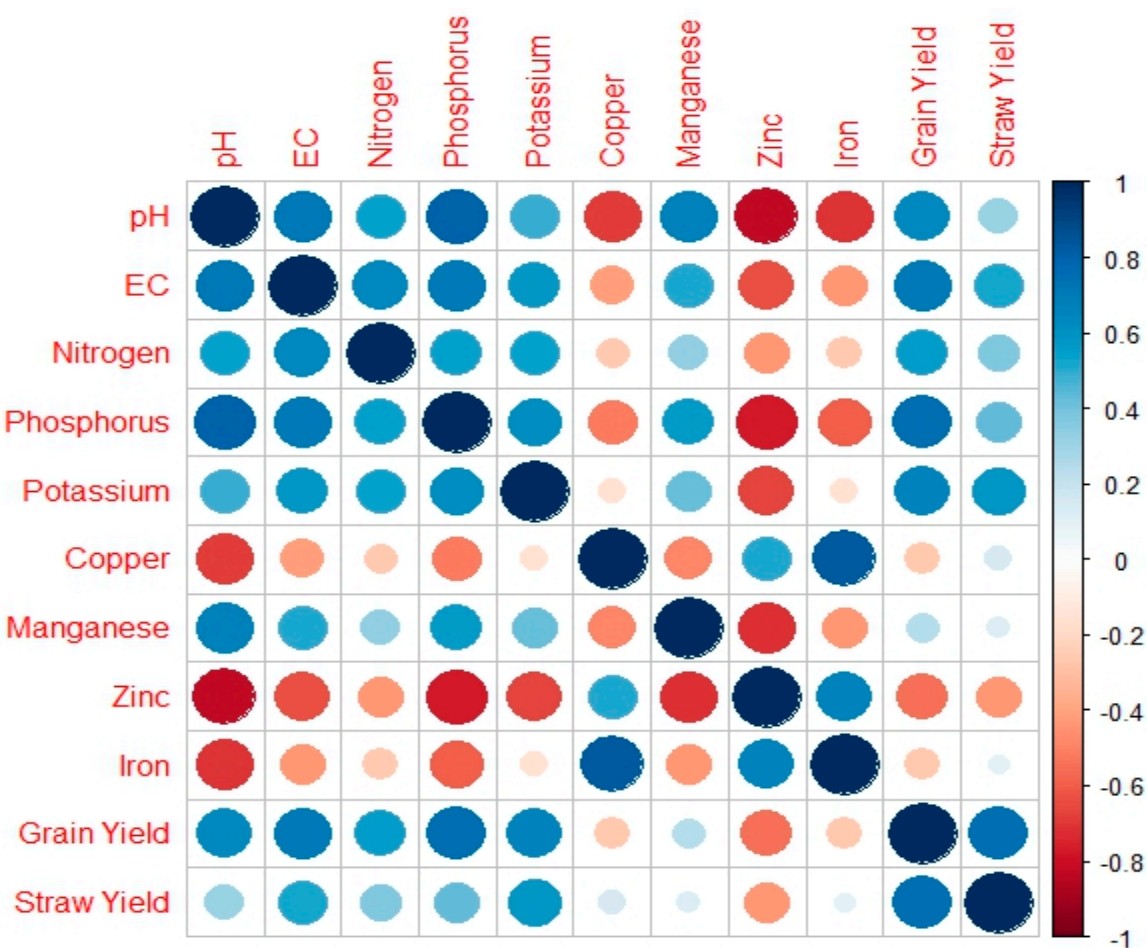

**Figure 4.** Correlation plots for different soil properties: blue color corresponds to (+) positive interaction, red color corresponds to (−) negative interaction, white color corresponds to neutral interaction among variables.

## 4. Conclusions

In view of the very high rate of fixation of P fertilizers applied to soil, it may be concluded, based on the above-mentioned findings, that the application of nano-P fertilizer, with a combination of different amounts of phosphatic fertilizers, showed improved results not only in yield, but also in chemical properties of post-harvest soil. The application of 100% NK + 75% P + 2 foliar sprays of nano-P at the tillering and panicle initiation stages ($T_4$) proved more effective for achieving higher growth, yield and yield attributing properties, and saved DAP if two foliar sprays of nano-P were applied at the tillering and panicle initiation stages as a form of phosphorus. Additionally, it can be a suitable substitute for 25% recommendation dose P for wheat crop cultivation under semi-arid climate. Further research trials need to be carried out to learn more about the efficacy of foliar application of nano-P.

**Author Contributions:** Conceptualization: A.P. (Anuj Poudel) and S.K.S.; methodology and visualization: A.P. (Anuj Poudel) and S.K.S.; software: S.S.J., A.P. (Abhik Patra) and A.P. (Astha Pandey); formal analysis: A.P. (Abhik Patra) and S.K.S.; investigation: A.P. (Anuj Poudel) and S.K.S.; writing—original draft preparation: A.P. (Anuj Poudel), S.K.S., S.S.J. and A.P. (Abhik Patra); writing—review & editing: S.K.S., S.S.J., A.P. (Abhik Patra) and R.J.-B. All authors have read and agreed to the published version of the manuscript.

**Funding:** This research has received no external funding.

**Institutional Review Board Statement:** Not applicable.

**Informed Consent Statement:** Not applicable.

**Data Availability Statement:** The data and materials will be made available by the corresponding author(s) upon reasonable request.

**Conflicts of Interest:** The authors declare no conflict of interest.

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
