# Peer review of "Effect of Nano-Phosphorus Formulation on Growth, Yield and Nutritional Quality of Wheat under Semi-Arid Climate"

_agronomy, doi:10.3390/agronomy13030768_

Round 1

Reviewer 1 Report

Dear Authors,

The topic of your manuscript is very interesting and fits well in the Agronomy journal. Nanofertilizers seems to be very promising strategy for the future. However, I have some suggestions, which should be considered before I agree with publication of your manuscript. I hope that changes will take you not much time and work.

General comments for whole manuscript:

NP abbreviation

I suggest to change the abbreviation for nano-phosphorus on Nano-P. The abbreviation NP resembles more Nitrogen-phosphorus fertilizing than nano-phosphorus application and is, therefore, confusing.

Rounding

I prefer the following numbers rounding system: if you have numbers higher than 100 it is not necessary to write them as 100.123, so round them to full numbers only. Numbers higher than 10 and lower than 100 should be then rounded to 10.1 format; numbers >1 and <10 to 1.12 format; numbers between 0 and 1 to 0.123 format. It is not necessary to change, but I think it will improve the overall transparency of your manuscript and tables.

Comments to the individual Chapters

Introduction

Lines 46-57: I suggest to delete the passage starting with "As a result.... (L46) and finishing with t "...to the grains" (L58) - it is commonly known and not directly related to your research. You are not investigating ATP or RNA contents etc.

Line 73: Unify the abbreviation - you use NURE, but in the remaining text only NUE (L85). On the line is the same abbreviation explained second time.

Line 99-100: I recommend the "internationalize" the costs and recalculate the Rs. to USD. Or maybe better - Express the average costs in % - for Nano-P (average price - Rs.374) as 100%; the average costs for SSP (Rs.560) as 150% and the average costs for DAP (Rs.1750) as 468%.

Line 128:  -1 should be in upper index

Materials and methods

Lines 160-161: delete "during the Rabi season of years 2020-2021" - you have it already at the lines 148-149.

Line 165: I recommend to re-calculate the oxide forms of nutrients to their pure forms in whole manuscript, e.g. P2O5 to P or K2O to K. I know that the oxide forms are commonly used, but I do not agree with this, because i) the fertilizers often do not contain these oxide forms (e.g., potassium muriate) and ii) the origin of mentioning these forms is probably only economical (not scientific), where by the using of oxide form it seems (only optically) that there is more nutrients. This is only recommendation not necessary to accept.

Table 1: The symbol * should be explained under the table -> * RDF - recommended dose of fertilizers.

Lines 195-196: 1) Double dot; 2) the citation should be in numeric form; 3) authors are not cited in the references.

Lines 197-199 and whole manuscript. I recommend to express the N, P and K contents in mg kg-1, which is more commonly used. Or at least, include the value of soil bulk density to make for reader possible of recalculation of kg ha-1 to mg kg-1.

Does the used fertilizer have some certified stability of the size of the nanoparticles?

Results

I suggest to rename this chapter on "Results and discussion"

Line 210: change "Table 2 that" on "Table 2 shows that"

Lines 245-246: I recommend to delete the name of subchapter 3.3. and rename the chapter 3.2. (L229) on "Yield parameters"

Figure 2. End the description of figure 2 with "... yield of wheat (T1=100%)”.

Line 270: maybe better "presented" than "furnished". Furthermore, I suggest to use present tense at the start of this sentence "... in Table 3 are depicting..."

Line 339: I suggest to change the name of chapter on "Nutrients uptake"

Line 364: Add here the chapter 3.7. Postharvest soil properties

Lines 375-378: The bioavailability of micronutrients is usually closely related to pH value. It is commonly known that especially nitrogen fertilizers are significantly influencing the soil pH. My question is, if you made the pH analysis in the postharvest samples? It can bring the new insight to the micronutrients behavior. If you have not the pH values, please try to discuss this possibility.

After considering my suggestions. I will be glad to recommend publish your manuscript in Agronomy Basel.

Yours sincerely

Reviewer

Author Response

Authors’ Response to the reviewer 1 comments: 

NP abbreviation

I suggest to change the abbreviation for nano-phosphorus on Nano-P. The abbreviation NP resembles more Nitrogen-phosphorus fertilizing than nano-phosphorus application and is, therefore,confusing.

Authors’ Response: Thank you very much for your suggestion. We have changed the nano-phosphorus (NP) abbreviation  into the Nano-P whole manuscript.

Rounding

I prefer the following numbers rounding system: if you havenumbers higher than 100 it is not necessary to write them as 100.123, so round them to full numbers only. Numbers higher than 10 and lower than 100 should be then rounded to 10.1 format;numbers >1 and <10 to 1.12 format; numbers between 0 and 1 to 0.123 format. It is not necessary to change, but I think it will improve the overall transparency of your manuscript and tables.

Authors’ Response: We have made rounding system in whole revised manuscript.

Comments to the individual Chapters

Introduction

Lines 46-57: I suggest to delete the passage starting with "As a result.... (L46) and finishing with t "...to the grains" (L58) - it is commonly known and not directly related to your research. You are not investigating ATP or RNA contents etc.

Authors’ Response: corrected it as per your comments.

Line 73: Unify the abbreviation - you use NURE, but in the remaining text only NUE (L85). On the line is the same abbreviation explained second time.

Authors’ Response: We have made corrected in whole manuscript in revised MS.

Line 99-100: I recommend the "internationalize" the costs and recalculate the Rs. to USD. Or maybe better - Express the average costs in % - for Nano-P (average price - Rs. 374) as 100%; the average costs for SSP (Rs. 560) as 150% and the average costs for DAP (Rs. 1750) as 468%.

Authors’ Response: corrected it as per your comments.

Line 128: -1 should be in upper index

Authors’ Response: Thank you very much for your suggestion. We have made superscript.

Materials and methods

Lines 160-161: delete "during the Rabi season of years 2020-2021"- you have it already at the lines 148-149.

Authors’ Response: corrected it as per your comments.

Line 165: I recommend to re-calculate the oxide forms of nutrients to their pure forms in whole manuscript, e.g. P2O5 to P or K2O to K. I know that the oxide forms are commonly used, but I do not agree with this, because i) the fertilizers often do not contain these oxide forms (e.g., potassium muriate) and ii) the origin of mentioning these forms is probably only economical (not scientific), where by the using of oxide form it seems (only optically) that there is more nutrients. This is only recommendation not necessary to accept.

Authors’ Response: Thank you for suggestion. We have made the suggested correction in updated MS.

Table 1: The symbol * should be explained under the table -> * RDF- recommended dose of fertilizers.

Authors’ Response: We have modified the text according to your suggestion

Lines 195-196: 1) Double dot; 2) the citation should be in numericform; 3) authors are not cited in the references.

Authors’ Response: Thank you for suggestion. We have made the suggested correction in updated MS.

Lines 197-199 and whole manuscript. I recommend to express the N, P and K contents in mg kg, which is more commonly used. Or at least, include the value of soil bulk density to make for reader possible of recalculation of kg ha to mg kg.

 Authors’ Response: Thank you for suggestion. We have modified N, P and K contents in mg kg-1    

Does the used fertilizer have some certified stability of the size of the nanoparticles?

Authors’ Response: Yes, the fertilizer have more stability at the size of the nanoparticles

Results

I suggest to rename this chapter on "Results and discussion"

Authors’ Response: We have rename the text according to your suggestion.

Line 210: change "Table 2 that" on "Table 2 shows that"

Authors’ Response: Thank you for suggestion and correction has been incorporated.

Lines 245-246: I recommend to delete the name of subchapter 3.3.and rename the chapter 3.2. (L229) on "Yield parameters"

Authors’ Response: The necessary correction has been made.

Figure 2. End the description of figure 2 with "... yield of wheat(T1=100%)”.

Authors’ Response: Thank you for suggestion. We have made the suggested correction in updated figure.

Line 270: maybe better "presented" than "furnished". Furthermore, Isuggest to use present tense at the start of this sentence "... in Table 3 are depicting..."

Authors’ Response: The suggested correction has been incorporated.

Line 339: I suggest to change the name of chapter on "Nutrients uptake"

Authors’ Response: The suggested correction has been incorporated.

Line 364: Add here the chapter 3.7. Postharvest soil properties

Authors’ Response: The suggested correction has been incorporated.

Lines 375-378: The bioavailability of micronutrients is usually closelyrelated to pH value. It is commonly known that especially nitrogen fertilizers are significantly influencing the soil pH. My question is, if you made the pH analysis in the post harvest samples? It can bring the new insight to the micronutrients behavior. If you have not the pH values, please try to discuss this possibility.

Authors’ Response: The suggested data has been added in the revised MS.

After considering my suggestions. I will be glad to recommend publish your manuscript in Agronomy Basel.

Reviewer 2 Report

Dear authors,

the topic of the work is very interesting to the scientific community. The experiment was set up and conducted well. A lot of effort was put into the whole work.

However, the writing style is not good. In your paper you just copied (rewrite) the results from the table (figures) with a few quotes. The discussion in the paper is very weak. The statistics are very simple, and you only have one year of research. The T4 treatment proved to be the best, but it was not explained why.

For the above reasons, I believe that you need to make an extra effort to paper published.

All suggestions are attached.

Best regards

Author Response

Dear Sir, all the suggestion made in revised manuscript

Reviewer 3 Report

Comments, Suggestion and Question.

Ø  Line 18. Kindly explain and clarify the meaning.

Ø  Line 20; applying 100% NK. How much N and how much K in this 100%.

Ø  Kindly remove the extra commas throughout the manuscript. Seriously focus on this issue.

Ø  Why you use different level of DAP?

Ø  Line 25; kindly find appropriate word instead of administered.

Ø  At the end of the Abstract, there should be a summary sentence about the practical significance of the results obtained, their practical consequences, and/or some advice or suggestions.

Ø  Line 71-74. The sentence is too long. Rewrite it.

Ø  What is smart delivery system? Explain it in introduction part?

Ø  Line 140-144; improve it and write the objective and hypothesis of this study?

Ø  What is the mean annual precipitation and the mean air temperature ranges?

Ø  What are the basic properties of soil before the experiment? Write the detail of soil in manuscript.

Ø  What is the innovation of this study?

Ø  Unfortunately, the quality of the picture in Figure 1is extremely low. I have tried to zoom it, but the inscription are unreadable. Kindly check it.

Ø  Why you use HD-2967 variety of wheat. Any characteristic. Why don’t use other varieties.

Ø  December to April is short time for wheat crop. Kindly check it again.

Ø  Why you apply 200 ml acre Nano-P. Give specific reason.

Ø  Line 175-176. Its rice or wheat. Basic mistake.

Ø  Which instrument is used to analyze available nitrogen available phosphorus, available potassium and cationic micronutrients (Cu, Mn, 189 Zn, Fe).

Ø  Why you use Duncan's multiple range test. Statistical analysis was not sufficiently discussed. Without statistics and research, the article does not contribute to any knowledge development.

Ø  Straw yield or straw biomass. Clarify it. You did not discuss in method material about biological yield and test wight.

Author Response

Authors’ Response to Reviewer # 3 comments

 Comments, Suggestion and Question.

Line 18. Kindly explain and clarify the meaning.

Authors’ Response: Thank you for suggestion. The suggested has been added in the revised MS.

Line 20; applying 100% NK. How much N and how much K in this 100%.

Authors’ Response: 100% NK means 120 kg N and 60 kg K.

Kindly remove the extra commas throughout the manuscript. Seriously focus on this issue.

Authors’ Response: Thank you for suggestion and removed the extra commas throughout the manuscript in the revised MS.

Why you use different level of DAP?

Authors’ Response: Increaing the frequency of nano-P increases the P supply to the plants and subsequently reducing the doses of DAP decreases the excess P supply to the crop as well as cost of cultivation. In this experiment we mainly focused on how nano-P supplemented with DAP fulfil the P requirement of the crop.

Line 25; kindly find appropriate word instead of administered.

Authors’ Response: The suggested correction has been incorporated.

At the end of the Abstract, there should be a summary sentence about the practical significance of the results obtained, their practical consequences, and/or some advice or suggestions.

Authors’ Response: Thank you for given this suggestion and all the suggestion  has been added in the revised Manuscript.

Line 71-74. The sentence is too long. Rewrite it.

Authors’ Response: Thank you for given this suggestion and we rewrote the sentences in revised manuscript.

What is smart delivery system? Explain it in introduction part?

Authors’ Response: It has been explained in the Introduction section.

Line 140-144; improve it and write the objective and hypothesis of this study?

Authors’ Response: The suggested correction has been incorporated.

What is the mean annual precipitation and the mean air temperature ranges?

Authors’ Response: Mean annual precipitation of the experimentalsite is 1110 mm and mean air temperature is 30.5oC.

What are the basic properties of soil before the experiment? Write the detail of soil in manuscript.

Authors’ Response: Thank you for given this suggestion and the basic properties in mentioned in revised manuscript.

What is the innovation of this study?

Authors’ Response: The innovation of the study incorporated in the Introduction section.

Unfortunately, the quality of the picture in Figure 1is extremely low. I have tried to zoom it, but the inscription are unreadable. Kindly check it.

Authors’ Response: Thank you for suggestion and we made the high resolution figure and added in revised manuscript

Why you use HD-2967 variety of wheat. Any characteristic. Why don’t use other varieties.

Authors’ Response: HD2967 Variety is very popular variety and its short duration with resistant all the three rust diease. 

December to April is short time for wheat crop. Kindly check it again.

Authors’ Response: The test crop var HD2967 has been matured within 125-135 days.

Why you apply 200 ml acre Nano-P. Give specific reason.

Authors’ Response:  Nano- P is developed by the IFFICO  and is recommedation of 500 ml per hacter for faremer so  as per the recommanadtion we calcuted the dose of nano-P.

Line 175-176. Its rice or wheat. Basic mistake.

Authors’ Response: We apologize for this and  this was wheat crop

Which instrument is used to analyze available nitrogen available phosphorus, available potassium and cationic micronutrients (Cu, Mn, 189 Zn, Fe).

Authors’ Response: Thank you for given this suggestion and all the instrument added that used to available nitrogen available phosphorus, available potassium and cationic micronutrients (Cu, Mn, 189 Zn, Fe). in revised manuscript.

Why you use Duncan's multiple range test. Statistical analysis was not sufficiently discussed. Without statistics and research, the article does not contribute to any knowledge development.

Authors’ Response: All the results were elaborated based on Duncan’s multiple range test (DMRT) in tables and figures.

Straw yield or straw biomass. Clarify it. You did not discuss in method material about biological yield and test wight.

Authors’ Response: Thank you for given this suggestion and  it clarify in revisied manuscript with added in method and material.

Reviewer 4 Report

Abstract

- The background of abstract should be revised.

- The experimental factors and treatments must be stated accurately.

- Line 21: Please add the values.

- What is the best recommendation for wheat growers?

Introduction

- This section is too long. Please delete non-important details.

- Lines 35-26: references?

- justify the novelty in introduction and discussion.

- The objectives and hypothesis should be added.

Materials and methods

- Line 154: Please add the soil physical and chemical properties of experimental area.

- In nano-P, why choose the dose of 200 ml acre-1?

- Please add more details about nano-P used in this experiment. The SEM, TEM images, nanoparticles size, etc.

Results and discussion

- Please add increasing or decreasing percentage.

- The discussion is missed.

- In this section the authors reported the obtained results of this study and after that compared the results with similar previously published studies. This is not a review article. The authors should be discussed clearly why the measured traits increase or decrease with under experimental treatments affects.

- Tables and Figures must be self-explanatory. That is, all abbreviations used must be clearly explained in the captions

- Lines 238-244: The similar results of previous studies should be deleted.

- Line 298: 308: This section should be deleted and add the main reasons for increasing or decreasing measured traits.

Conclusion

- This section is repetitive and should be rewritten.

- Please make sure your conclusions' section underscores the scientific value-added of your paper, and/or the applicability of your findings/results. Highlight the novelty of your study.

Author Response

Authors’ Response to Reviewer # 4 comments

CommentsandSuggestionsfor Authors

Abstract

- The background of abstract should be revised.

Authors’ Response: Thank you for given this suggestion and we revised background of abstract in revised Manuscript.

- The experimental factors and treatments must be statedaccurately.

Authors’ Response: The suggested correction has been incorporated

- Line 21: Please add the values.

Authors’ Response: The suggested correction has been incorporated

- What is the best recommendation for wheat growers?

Authors’ Response: The suggested correction has been incorporated in the Conclusion section i.e., application of 100% NK + 75% P + 2 foliar sprays of nano-P at tillering and panicle initiation stage.

Introduction

This section is too long. Please delete non-important details.

Authors’ Response: The suggested correction has been incorporated.

- Lines 35-26: references?

Authors’ Response: The suggested correction has been incorporated and added proper citation in revised manuscript.

- justify the novelty in introduction and discussion.

Authors’ Response: The suggested correction has been incorporated.

Valuable comment indeed. The “Introduction” section has been revised rigorously and now incorporates the novelty of this review work at the last paragraph of the “Introduction” section.

- The objectives and hypothesis should be added.

Authors’ Response: The suggested correction has been incorporated

Materials and methods

- Line 154: Please add the soil physical and chemical properties of experimental area.

Authors’ Response: Initial soil properties has been given in section 2.1 Study area

- In nano-P, why choose the dose of 200 ml acre-1?

Authors’ Response: Nano- P is developed by IFFCO and is recommedation of 500 ml per hacter for faremer so  as per the recommanadtion we calcuted the dose of nano-P.

- Please add more details about nano-P used in this experiment. The SEM, TEM images, nanoparticles size, etc.

Authors’ Response: We will add this data in subsequent research papers.

Results and discussion

- Please add increasing or decreasing percentage.

Authors’ Response: The suggested correction has been incorporated.

- The discussion is missed.

Authors’ Response: The suggested correction has been incorporated.

- In this section the authors reported the obtained results of thisstudy and after that compared the results with similar previouslypublished studies. This is not a review article. The authors should be discussed clearly why the measured traits increase or decrease withunder experimental treatments affects.

Authors’ Response: The suggested correction has been incorporated.

- Tables and Figures must be self-explanatory. That is, allabbreviations used must be clearly explained in the captions

Authors’ Response: The suggested correction has been incorporated

- Lines 238-244: The similar results of previous studies should bedeleted.

Authors’ Response: The suggested correction has been incorporated.

- Line 298: 308: This section should be deleted and add the mainreasons for increasing or decreasing measured traits.

Authors’ Response: The suggested correction has been incorporated.

Conclusion

- This section is repetitive and should be rewritten.

The suggested correction has been incorporated.

- Please make sure your conclusions' section underscores thescientific value-added of your paper, and/or the applicability of yourfindings/results. Highlight the novelty of your study.

Authors’ Response: The suggested correction has been incorporated

Again, the authors are very much thankful to the anonymous reviewers for the careful review and constructive critical comments on our manuscript. We are confident that the revised version has been much improved in its academic quality and will be accepted for publication in its present form.

Best regards and appreciations,

Corresponding authors:

Round 2

Reviewer 1 Report

Dear Authors,

Thanks for the corrections according to my comments. Now it is my pleasure to recommend publish your manuscript in Agronomy.

Yours Sincerely

Reviewer

Author Response

Authors’ Response to Reviewer # 1 (Round 2) comments

Comments and Suggestions for Authors

Dear Authors,

Thanks for the corrections according to my comments. Now it is my pleasure to recommend publish your manuscript in Agronomy.

Yours Sincerely

Reviewer

Authors’ Response: Thank you very much sir for appreciating our work

Best regards and appreciations,

Reviewer 4 Report

I have read the revised manuscript (agronomy-1979325).  Titled: Effect of Nano-Phosphorus Formulation on Growth, Yield and Nutritional Quality of Wheat under Semi-arid Climate for publication of agronomy MDPI. The authors addressed the questions and that reviewers raised the issue in the review of the original manuscript. I satisfy the author’s revisions throughout the paper. The author well-addresses all the questions and quarries in this manuscript. Before accepting this manuscript if there is anything needed to be revised by the authors, especially problems and questions in the text of the article, and journal format, I request this manuscript is currently in “major Revision” and the author may correct the problems and questions the author may improve in this stage. Thank you

Author Response

Authors’ Response to Reviewer # 4 (Round 2) comments

Please apply journal format in all sections

Authors’ Response: Thank you very much sir for this suggestion we have made all the section as per the Agronomy Journal format in revised manuscript.

Alphabetical order is not respected Please fix it

Authors’ Response: Suggestion in incorporated and changed the all keyword as per your suggestion sir. Example

Randomized complete block design (RCBD) please add

Authors’ Response: Randomized block design (RBD) is changed into the Randomized complete block design (RCBD) in revised manuscript

Please add how to synthesize and prepare your nanoparticles

Authors’ Response: We regret to inform you that the nono-P synthesis technique will not be disclosed in the study as it was synthesis by public-privet firm partnership. Moreover, we want to file a patent for this protocol. Hope you understand our situation.

Please add your nanoparticle size

Please add SEM and TEM image

Authors’ Response: Thank you very much sir for the wonderful suggestion but, SEM and TEM image characterization instruments of our department are not working properly and We are mentioning here that the size of the Nano-P particle is small (10 to 30 nanometers). and we have already sent the sample to another referral laboratory for the analysis of SEM and TEM.

Reference??

Authors’ Response: Suggestion in incorporated in revised manuscript and added the reference

What test was used for the data to be normal? please add

Authors’ Response: Dear sir, we not used any tool or test for data normal. For data normalization we used mean of three replicated data. And more over for data visualization we use the statistical analysis and depicted CD, SEm± and DMRT test.

Why yield traits not reviewed in Pearson correlation analysis?

Authors’ Response: We did Pearson correlation analysis to examine the all yield trait and the results were inconclusive. As a result, we did not include all the yield traits however, in revised manuscript we include the grain and straw yield data in person correlation analysis.

Please add ANOVA table (mean square value) or p-value for each variables

Authors’ Response: Thank you for your suggestions, Sir. We mentioned the critical differences value, significant differences at the 5% level, standard error of mean value, and Duncan's test in each table. As a result, we are not mentioning the ANOVA table in tables, which affects the quality and readability of the tables, and in most other publications, the ANOVA table is not included.

It is wrong to be corrected

Authors’ Response: Suggestion in incorporated and corrected the value in revised manuscript.

Please add: Different letters indicate significant differences at the 5% level according to Duncan's test

Authors’ Response: Thank you very much sir for this suggestion we have add different letters indicate significant differences at the 5% level according to Duncan's test in all the table in revised manuscript

P values is not clear, please correct the figure

Authors’ Response: Suggestion in incorporated and corrected in revised manuscript.

Please add abbreviation:

Food security: Food Secur.

Please add journal abbreviation for all reference

Authors’ Response: Thank you sir for the suggestions we have made all the references as per the Journal formatting style in the revised manuscript

Please add DOI for all reference

Authors’ Response: Dear sir we added the DOI number of all these papers that have the DOI number on Google Scholar in Revised manuscript.

Again, the authors are very much thankful to the anonymous reviewer for the careful review and constructive critical comments on our manuscript. We are confident that the revised version has been much improved in its academic quality and will be accepted for publication in its present form.

Best regards and appreciations,
